# Euclidean distance compression via deep random features

**Brett Leroux**
Department of Mathematics
University of California, Davis
Davis, CA 95616
lerouxbew@gmail.com

**Luis Rademacher**
Department of Mathematics
University of California, Davis
Davis, CA 95616
lrademac@ucdavis.edu

## Abstract

Motivated by the problem of compressing point sets into as few bits as possible while maintaining information about approximate distances between points, we construct random nonlinear maps $\varphi_\ell$ that compress point sets in the following way. For a point set $S$, the map $\varphi_\ell : \mathbb{R}^d \to N^{-1/2}\{-1, 1\}^N$ has the property that storing $\varphi_\ell(S)$ (a *sketch* of $S$) allows one to report squared distances between points up to some multiplicative $(1 \pm \epsilon)$ error with high probability. The maps $\varphi_\ell$ are the $\ell$-fold composition of a certain type of random feature mapping.

Compared to existing techniques, our maps offer several advantages. The standard method for compressing point sets by random mappings relies on the Johnson-Lindenstrauss lemma and involves compressing point sets with a random linear map. The main advantage of our maps $\varphi_\ell$ over random linear maps is that ours map point sets directly into the discrete cube $N^{-1/2}\{-1, 1\}^N$ and so there is no additional step needed to convert the sketch to bits. For some range of parameters, our maps $\varphi_\ell$ produce sketches using fewer bits of storage space. We validate the method with experiments, including an application to nearest neighbor search.

## 1 Introduction

Random projection is a commonly used method to lower the dimension of a set of points while maintaining important properties of the data [30]. The random projection method involves mapping a high-dimensional set of points in $\mathbb{R}^d$ to a lower dimensional subspace by some random projection matrix in such a way that the pairwise distances and inner products between points are approximately preserved. The random projection method has many applications to data analysis and a variety of prominent algorithms [11, 17, 5, 8] including nearest neighbor search [13, 12].

The theoretical foundation of random projection is the Johnson-Lindenstrauss lemma which states that a random orthogonal projection to a lower dimensional subspace has the property of preserving pairwise distances and inner products [16]. Later it was observed [9, 13] that one can alternatively take the projection matrix to be a matrix with i.i.d. Gaussian $N(0, 1)$ entries.

**Lemma 1** ([9, 13, 30]). *Let each entry of an $d \times k$ matrix $R$ be chosen independently from $N(0, 1)$. Let $v = \frac{1}{\sqrt{k}} R^T u$ for $u \in \mathbb{R}^d$. Then for any $\epsilon > 0$, $\mathbb{P}\big(\big|\|v\|^2 - \|u\|^2\big| \geq \epsilon\|u\|^2\big) < 2e^{-(\epsilon^2 - \epsilon^3)k/4}$.*

A corollary of the above lemma is that if an arbitrary set of $n$ points in $\mathbb{R}^d$ is mapped by the random projection matrix $R$ to $\mathbb{R}^k$ where $k = \Theta(\epsilon^{-2} \log n)$, then the squared distances between pairs of points are distorted by a factor of at most $(1 \pm \epsilon)$ with high probability. The projected points are thus a lower dimensional representation of the original point set and this lowering of the dimension offers two main advantages. The first is that algorithms that were originally intended to be performed on the original point set can now instead be performed on the lower dimensional points.

38th Conference on Neural Information Processing Systems (NeurIPS 2024).

The second main advantage of the lower dimensional representation is a reduction in the cost of data storage. The Johnson-Lindenstrauss lemma shows that random projection of a point set $S$ produces a data structure (called a *sketch*) that can be used to report the squared distances between pairs of points up to a multiplicative $(1 \pm \epsilon)$ error. The size of the sketch of course depends on both $|S|$ and $\epsilon$. From this viewpoint, it is natural to ask what is the minimum number of bits of such a sketch? The Johnson-Lindenstrauss lemma gives an upper bound on the size of such a sketch as follows: Any set of $n$ points $S$ can be projected to $\mathbb{R}^k$ where $k = \Theta(\epsilon^{-2} \log n)$ while distorting pairwise squared distances by a factor of at most $(1 \pm \epsilon)$. The projected data points are real-valued, and thus the projected data points needs to be encoded into bits in such a way that guarantees squared distances are preserved. One way to convert to bits is to use an epsilon-net for the unit ball in $\mathbb{R}^k$: In order to preserve squared distances up to a multiplicative $(1 \pm \epsilon)$ error, it suffices to preserve squared distances up to an additive $m^2 \epsilon$ error where $m$ is the minimum distance between pairs of points in $S$. By identifying each projected point with the closest point in an $m^2 \epsilon$-net, we can produce a sketch with $\Theta\big(n\epsilon^{-2} \log n \log(1/m^2\epsilon)\big)$ bits.[1]

While the Johnson-Lindenstrauss lemma shows that efficient sketches can be obtained by mapping the points to a lower dimensional space with a random linear mapping (the projection), it is natural to ask if there are other types of random maps (in particular, possibly nonlinear maps) which are able to produce sketches with a smaller number of bits. Our main result shows that this is possible in certain cases by using the composition of random feature maps. We state our main result first for sets of points contained in the unit sphere $S^{d-1}$ and at the end of this section we include the extension to subsets of the unit ball.

**Theorem 2.** *Let $S \subset S^{d-1}$ with $|S| = n \geq 2$. Let $m = \min_{x,y \in S, x \neq y} \|x - y\|$ and $\ell = \lceil \log_2 \log_2 \frac{4}{m} \rceil \geq 1$. Let $\epsilon > 0$ and assume that $\epsilon < \min_{x,y \in S, x \neq y} 1 - |\langle x, y \rangle|$. Then the random map $\varphi_\ell : S^{d-1} \to \frac{1}{\sqrt{N}}\{-1, 1\}^N$ with $N = \Theta\big(\frac{\log n}{\epsilon^2}(\log \frac{1}{m})^{2 \log_2(\pi/\sqrt{2})}\big)$ (defined in the proof of Theorem 5 and independent of $S$ except through parameters $n, d, m$ and $\min_{x,y \in S, x \neq y} 1 - |\langle x, y \rangle|$) satisfies the following with probability at least $(1 - \frac{2}{n})^\ell$: $\varphi_\ell(S)$ is a sketch of $S$ that allows one to recover all squared distances between pairs of points in $S$ up to a multiplicative $(1 \pm \epsilon)$ error. The number of bits of the sketch is $\Theta\big(\frac{n \log n}{\epsilon^2}(\log \frac{1}{m})^{2 \log_2(\pi/\sqrt{2})}\big)$.*

The proof of Theorem 2 is explained in Section 3 and the final details of the proof are in Appendix A. We explain how the map $\varphi_\ell$ is constructed in the next subsection and the role of the parameter $\ell$ is discussed in Section 1.2. The main advantage of the map $\varphi_\ell$ is that it maps the point set $S$ directly into the discrete cube and thus there is no need to convert the sketch to bits after performing the random mapping. Furthermore, the map $\varphi_\ell$ produces sketches with asymptotically fewer bits than those obtained using the Johnson-Lindenstrauss lemma if $(\log(\frac{1}{m}))^{2 \log_2(\pi/\sqrt{2})} = o\big(\log(\frac{1}{m^2\epsilon})\big)$. This is equivalent to the condition that $(\log(\frac{1}{m}))^{2 \log_2(\pi/\sqrt{2})} = o\big(\log(\frac{1}{\epsilon})\big)$.

Furthermore, the sketch $\varphi_\ell(S)$ has the desired properties "with high probability". The probability that the sketch succeeds is $(1 - 2/n)^\ell$ and we claim that this quantity approaches one for all useful choices of the parameters: First of all, we recall that in all applications of Theorem 2, we should take $n = \Omega(d)$. We also take $\epsilon = \Omega(d^{-1/2})$ (ignoring logarithmic factors) because otherwise the target dimension $N$ is larger than $d$ and then a sketch based on an $\epsilon$-net (as above, but without random projection) for $S$ would be better, by having bitlength $nd$ (up to logarithmic factors). The assumption that $\epsilon < 1 - |\langle x, y \rangle|$ for all $x, y, x \neq y$ means that $\epsilon < m^2$ and so $1/m < \epsilon^{-1/2} = O(d^{1/4})$. This means that $\ell = O(\log_2 \log_2 d)$ and so $(1 - 2/n)^\ell = \Omega(1 - 2/d)^\ell$ approaches one.

We remark that it might be possible to replace the assumption in Theorem 2 that $\epsilon < 1 - |\langle x, y \rangle|$ for all $x, y \in S$, $x \neq y$ by the weaker assumption that $\epsilon < 1 - \langle x, y \rangle = \frac{1}{2}\|x - y\|^2$. The reason that an assumption of this sort is necessary is that, because we are trying to produce sketches which allow recovery of squared distances, pairs of points with very small distance are difficult to deal with. As a result, we need to assume for technical reasons that the accuracy parameter $\epsilon$ is smaller then (half) the minimum squared distance, i.e. $\epsilon < 1 - \langle x, y \rangle$. There is an inherent symmetry in the maps $\varphi_\ell$ that we use which makes it convenient to use the stronger assumption that $\epsilon < 1 - |\langle x, y \rangle|$.

---

[1] Since the points are in the unit ball, for any $\delta > 0$, approximating distances to within additive error $\delta/6$ gives approximation of squared distances to within additive error $\delta$. So the size of the epsilon net is, up to a constant, $(1/m^2\epsilon)^{\epsilon^{-2} \log n}$. This gives $\Theta\big(\log(1/m^2\epsilon)\epsilon^{-2} \log n\big)$ bits per point

We are able to extend our main result to deal with not only sets of points contained in the unit sphere, but also any set of points in the unit ball. For a point $x$ in the unit ball $B^d$ we use $\hat{x}$ to denote $x/\|x\|$.

**Theorem 3.** *Let $S \subset B^d$ with $|S| = n \geq 2$ and set $\rho = \min_{x \in S} \|x\|^2$. Let $m = \min_{x,y \in S, x \neq y} \|\hat{x} - \hat{y}\|$ and $\ell = \lceil \log_2 \log_2 \frac{4}{m} \rceil \geq 1$. Let $\epsilon > 0$ and assume that $\epsilon < \min_{x,y \in S, x \neq y} 1 - |\langle \hat{x}, \hat{y} \rangle|$. Then the random map $\varphi_\ell : S^{d-1} \to \frac{1}{\sqrt{N}}\{-1,1\}^N$ with $N = \Theta\left(\frac{\log n}{\epsilon^2}(\log \frac{1}{m})^{2\log_2(\pi/\sqrt{2})}\right)$ (defined in the proof of Theorem 5 and independent of $S$ except through parameters $n, d, m$ and $\min_{x,y \in S, x \neq y} 1 - |\langle \hat{x}, \hat{y} \rangle|$) satisfies the following with probability at least $(1 - \frac{2}{n})^\ell$: $\varphi_\ell(\hat{S})$ and the norm of each point in $S$ up to an additive $\pm \rho m^2 \epsilon/48$ error is a sketch of $S$ that allows one to recover all squared distances between pairs of points in $S$ up to a multiplicative $(1 \pm \epsilon)$ error. Moreover the number of bits of the sketch is $\Theta\left(\frac{n \log n}{\epsilon^2}\left(\log \frac{1}{m}\right)^{2\log_2(\pi/\sqrt{2})} + n \log \frac{1}{\rho m^2 \epsilon}\right)$.*

The proof of Theorem 2 is explained in Section 3 and the final details of the proof are in Appendix A. The number of bits of the sketches in the above theorem depend on the parameter $m$ which is the minimum distance between pairs of points after all points have been normalized to have unit norm. Thus, it is more complicated to compare the number of bits of our sketches to the sketches obtained using the Johnson-Lindenstrauss lemma because the Johnson-Lindenstrauss sketches do not rely on any normalization step. However, as in the case of point sets on $S^{d-1}$, there is a large family of point sets in $B^d$ for which our sketching technique produces sketches with a fewer number of bits.

## 1.1 The maps $\varphi_\ell$ and the recovery of $\|x - y\|^2$ by $\varphi_\ell(x)$ and $\varphi_\ell(y)$

In this section we summarize the construction of the maps $\varphi_\ell$ which are used in Theorem 2 and formally analyzed in Theorem 5 and how they can be used to recover squared distances between points. Let $f(t) = \frac{2}{\pi}\arcsin(t)$ and $g(t) = \sin(\frac{\pi t}{2})$. So $f : [-1,1] \to [-1,1]$ is the inverse of $g : [-1,1] \to [-1,1]$. For $\ell \in \mathbb{N}^+$, let $f_\ell$ be the function $f$ composed with itself $\ell$ times and similar for $g_\ell$. Notice that for any $\ell \in \mathbb{N}^+$, $f_\ell : [-1,1] \to [-1,1]$ is the inverse of $g_\ell : [-1,1] \to [-1,1]$.

For simplicity in the rest of the introduction we assume all points are normalized to be on the unit sphere $S^{d-1}$. We define the sign function as

$$\mathrm{sign}(t) = \begin{cases} 1 & \text{if } t \geq 0 \\ -1 & \text{if } t < 0. \end{cases}$$

The maps $\varphi_\ell$ will be defined as the composition of $\ell$ maps of the following form. Set $D \in \mathbb{N}^+$. Let $Z_i, 1 \leq i \leq D$ be i.i.d. standard Gaussian random vectors in $\mathbb{R}^d$. Let $\varphi^D : \mathbb{R}^d \to \mathbb{R}^D$ be defined by

$$\varphi^D(x)_i := D^{-1/2}\mathrm{sign}(\langle x, Z_i \rangle)$$

where $\varphi^D(x)_i$ is the $i$th coordinate of $\varphi^D(x)$.[2]

The maps $\varphi_\ell$ are now defined as the $\ell$-fold composition of maps of the type $\varphi^D$. That is, for some integers $D_1, D_2, \ldots, D_\ell$, we let $\varphi_1 : S^{d-1} \to D_1^{-1/2}\{-1,1\}^{D_1}$ be defined by $\varphi_1(x) = \varphi^D(x)$. For $j \in \{1, \ldots, \ell - 1\}$, we let $\varphi_{j+1}(x) = \varphi^{D_{j+1}}(\varphi_j(x))$. Therefore, the map $\varphi_\ell$ maps $S^{d-1}$ to $D_\ell^{-1/2}\{-1,1\}^{D_\ell}$. To avoid writing the double subscript we write the final dimension of the map as $D_\ell = N$. We remark that the map $\varphi_\ell$ can also be defined as a standard neural network with $\ell$ hidden layers using activation function $\mathrm{sign}(t)$ and having all weights be i.i.d. standard Gaussian random variables.

It was shown in [26] that for $x, y \in S^{d-1}$, $\mathbb{E}\,\mathrm{sign}(\langle x, Z_1 \rangle)\mathrm{sign}(\langle y, Z_1 \rangle) = f(\langle x, y \rangle)$. Since $\langle \varphi^D(x), \varphi^D(y) \rangle$ is a sum of $D$ independent copies of $D^{-1}\mathrm{sign}(\langle x, Z \rangle)\mathrm{sign}(\langle y, Z \rangle)$ we get that $\mathbb{E}\langle \varphi^D(x), \varphi^D(y) \rangle = f(\langle x, y \rangle)$.

Now we can explain how one recovers pairwise distances between points in $S$ from $\varphi_\ell(S)$. Let $S$ be a set of $n$ points in $S^{d-1} \subset \mathbb{R}^d$. As in Theorem 2, we map $S$ by $\varphi_\ell : S^{d-1} \to N^{-1/2}\{-1,1\}^N$. Here $\ell$ is some parameter which is chosen based on $S$ that is explained below and $N$ is chosen based on the desired $\epsilon$ error of the sketch. If the remaining integers $D_1, \ldots, D_{\ell-1}$ are chosen properly,

---

[2]We remark that the map $\varphi^D$ is often referred to as a *random feature map*, see [28, 7, 26].

then we show that $\langle \varphi_\ell(x), \varphi_\ell(y) \rangle$ is a good approximation of $f_\ell(\langle x, y \rangle)$ (Corollary 4). Since $g_\ell$ is the inverse of $f_\ell$ this implies that $g_\ell(\langle \varphi_\ell(x), \varphi_\ell(y) \rangle)$ should be a good approximation of $\langle x, y \rangle$. By the polarization identity, this implies that $2 - 2g_\ell(\langle \varphi_\ell(x), \varphi_\ell(y) \rangle)$ should be a good approximation of $\|x - y\|^2$. So recovering $\|x - y\|^2$ from $\varphi_\ell(S)$ simply involves calculating $2 - 2g_\ell(\langle \varphi_\ell(x), \varphi_\ell(y) \rangle)$. In the next section we explain why this mapping and recovery scheme leads to good error guarantees.

## 1.2 Intuition behind the construction

Now that we have defined the maps $\varphi_\ell$ we can explain the idea behind using maps of this form. The reason why this type of map is useful has to do with the behavior of the derivative of the function $g_\ell$ near $t = 1$. As previously mentioned, the map $\varphi_\ell$ has the property that for all $x, y \in S$,

$$|\langle \varphi_\ell(x), \varphi_\ell(y) \rangle - f_\ell(\langle x, y \rangle)| < \delta \tag{1}$$

for some $\delta$ depending on $N$. Now when we want to recover $\|x - y\|^2$ based on $\varphi_\ell(x)$ and $\varphi_\ell(y)$, by the polarization identity, we use $2 - 2g_\ell(\langle \varphi_\ell(x), \varphi_\ell(y) \rangle)$ as an estimate of $\|x - y\|^2$. The additive error of the approximation of $\|x - y\|^2$ by $2 - 2g_\ell(\langle \varphi_\ell(x), \varphi_\ell(y) \rangle)$ depends on the error in Eq. (1) as well as the derivative of $g_\ell$ near the point $f_\ell(\langle x, y \rangle)$. The function $g_\ell$ has the property that its derivative approaches zero as $t$ approaches one and so the additive error of the approximation of $\|x - y\|^2$ by $2 - 2g_\ell(\langle \varphi_\ell(x), \varphi_\ell(y) \rangle)$ gets smaller the closer $f_\ell(\langle x, y \rangle)$ (and thus $\langle x, y \rangle$) is to one, i.e., the closer $\|x - y\|^2$ is to zero. (This is quantified in Theorem 6, specifically the exponent approaching 2 in the additive error) The effect that this has is that we actually approximate $\|x - y\|^2$ up to a multiplicative error.

The role of the parameter $\ell$ in this construction is in controlling the rate at which $g_\ell'(t)$ approaches zero as $t$ approaches one. If there are pairs $x, y$ such that $\|x - y\|^2$ is very small then we need the derivative to approach zero very quickly. This can be done by increasing the parameter $\ell$; the rate at which $g_\ell'$ approaches zero is faster for larger values of $\ell$ (see Theorem 7 in Appendix B). It turns out that the correct choice of $\ell$ is $\lceil \log_2 \log_2 r \rceil$ where $r$ is approximately the reciprocal of the minimum distance between pairs of points in $S$ (see Theorem 2). Because of the connection with standard neural networks with $\ell$ hidden layers mentioned in Section 1.1, we refer to the parameter $\ell$ as the number of *layers* of the random mapping.

We remark that the function $g_\ell(t)$ has the property that its derivative can be as large as $\approx (\frac{\pi}{2})^\ell$ when $t$ is near zero (see Lemma 8 in Appendix B). The effect that this has is that our algorithm leads to worse approximation of $\|x - y\|^2$ when $\langle x, y \rangle$ is close to zero if $\ell$ is large. However, this loss in accuracy is made up for by increasing $N$ by only a relatively small amount and the gain in accuracy when $\langle x, y \rangle$ is close to one outweighs the loss in accuracy when $\langle x, y \rangle$ is close to zero.

## 1.3 Previous variations on random projection

Our compression method is similar to random projection in that they both involve compressing a set of points by randomly mapping it to a lower dimensional space. A number of other papers have also suggested variations on random projection where a different random mapping is used. In some, the random mapping is still linear. In others, they use a linear mapping followed by some quantization step. The main difference in our method is that it is more fundamentally non-linear due to the fact that it is a "deep" composition of non-linear maps.

One of the standard versions of random projection involves mapping points by a random Gaussian matrix. It was later shown that other types of random matrices work equally well. [1, 3, 30, 23]. In particular, the "binary" version of the Johnson-Lindenstrauss lemma due to [1] (where the entries of the matrix are all either $+1$ or $-1$) is particularly important for the following reason. As discussed in [15], an alternate way to convert sketches obtained by the Johnson-Lindenstrauss lemma to bits is possible if the points have bounded integer coordinates and one uses the binary variant of Johnson-Lindenstrauss lemma. This approach is somewhat incomparable to our setting because of the integrality assumption.

Another variation on the random projection technique is to apply a quantization step after the projection which further reduces the cost of storing the data points [6, 21, 24, 29, 4, 10, 18, 25, 32]. A particularly relevant version of quantization is "sign random projections" [6]. Sign random projections are the same as the 1-layer maps $\varphi_1$. They were used to estimate angles between points in

[6] and used to estimate inner products between points in [22]. Therefore, the main novelty of our technique is the idea of composing multiple such maps.

## 1.4   Distance compression beyond random mappings

Random mappings are of course not the only way to compress a data set. Here we compare our method to compression techniques that use methods other than random mappings. These methods tend to be more complicated algorithmically but as we explain below, can produce sketches with fewer bits.

Given a set of $n$ points in the unit ball in $\mathbb{R}^d$, what is the minimum number of bits of a sketch which allows one to recover all pairwise distances up to a multiplicative $(1 \pm \epsilon)$ error? As explained above, the Johnson-Lindenstrauss lemma shows that $O\big(\epsilon^{-2} n \log n \log(1/m^2 \epsilon)\big)$ bits suffice. However, this is not the optimal number of bits. It was recently shown in [14, 15] that if the points are contained in the unit ball and $m$ is the minimum distance between points, then $O\big(\epsilon^{-2} n \log n + n \log \log(1/m)\big)$ bits suffice. The additive error version of this question was answered in [2]. Previous to the result of [14, 15], the best known result was that $O\big(\epsilon^{-2} n \log n \log(1/m)\big)$ bits suffice [19]. These two results, however, use sketching techniques that differ from the sketches obtained by the random projection technique in a fundamental way. Random projection compresses the data set "point by point" in the sense that the compression process is applied to each point independently from the others. In contrast, the sketches in [15] and [19] must compress the entire data set simultaneously.

Another way of stating this distinction is that "point by point" methods (such as random projection) satisfy the requirements of the *one-way communication* version of this sketching problem while the methods used in [15] and [19] do not. In the *one-way communication* version of the sketching problem, Alice holds half of the data points and Bob holds the other half. Alice sends a message to Bob using as few bits as possible. Bob then must report distances between pairs of points where one point in the pair is known by Alice and the other by Bob. The one-way communication version of the sketching problem asks one to determine the minimum number of bits of Alice's message. It was shown in [27] that if the points are in the unit ball and the minimum distance is $m$, then $\Omega\big(\epsilon^{-2} n \log(n/\delta) \log(1/m)\big)$ bits are required for the one-way communication version of the problem if the sketch is required to be successful with probability at least $1 - \delta$.

Any sketching algorithm which compresses the data set point by point satisfies the requirements of the one-way communication variant of the sketching problem. We therefore know that sketching algorithms which compress the data set point by point cannot produce sketches with the optimal number of bits. However, there are several advantages to sketching algorithms of this sort. One advantage is that they are generally simpler and easier to implement. Another is that if one wants to add additional points to the data set, the entire sketching algorithm does not need to be re-run.

Our sketching algorithm from Theorem 2 also has the property that it compresses the data set point by point. Furthermore, the number of bits of our sketch almost matches the lower bound from [27]. The dominant term in the bound from Theorem 2 is $\Theta(\epsilon^{-2} n \log n (\log(1/m))^{2 \log_2(\pi/\sqrt{2})})$. Thus our number of bits matches the lower bound from [27] up to the power on the $\log(1/m)$ term. This motivates the question of whether some variation on our sketching technique can reduce this power.

## 1.5   Outline of the paper and notation

Section 2 contains the construction of the maps $\varphi_\ell$ and quantifies the error in the approximation of $f_\ell(\langle x, y \rangle)$ by $\langle \varphi_\ell(x), \varphi_\ell(y) \rangle$. Then in Section 3 we explain how the proof of Theorems 2 and 3 is completed. This amounts to showing how $\langle \varphi_\ell(x), \varphi_\ell(y) \rangle$ allows one to estimate $\|x - y\|^2$ and quantifying the error of the estimation. Some of the details are deferred to the appendix. Finally, Section 4 contains experimental results, including an application to nearest neighbor search.

For $x \in \mathbb{R}^d \setminus \{0\}$ we use $\hat{x}$ to denote $x/\|x\|$. For $x, y \in S^{d-1}$, the *polarization identity* states that $2 - 2\langle x, y \rangle = \|x - y\|^2$; for arbitrary $x, y \in \mathbb{R}^d$, it states that $\|x\|^2 + \|y\|^2 - 2\langle x, y \rangle = \|x - y\|^2$.

## 2   The construction of the maps $\varphi_\ell$

The purpose of this section is to prove the following result, Corollary 4, which shows a bound on the error of the approximation of $f_\ell(\langle x, y\rangle)$ by $\langle\varphi_\ell(x), \varphi_\ell(y)\rangle$ for all pairs $x, y$ in a set of $n$ points. The error in this approximation depends on the dimension $D_\ell$ of the image space of $\varphi_\ell$. In particular, we show how large $D_\ell$ needs to be in order to guarantee with high probability that $\langle\varphi_\ell(x), \varphi_\ell(y)\rangle$ is equal to $f_\ell(\langle x, y\rangle)$ up to some additive error $\delta$ for all pairs $x, y$.

We recall the definitions of the functions $f_\ell$ and $g_\ell$: Let $f(t) = \frac{2}{\pi}\arcsin(t)$ and $g(t) = \sin(\frac{\pi t}{2})$. For $\ell \in \mathbb{N}^+$, let $f_\ell$ be the function $f$ composed with itself $\ell$ times and similar for $g_\ell$. Notice that for any $\ell \in \mathbb{N}^+$, $f_\ell : [-1, 1] \to [-1, 1]$ is the inverse of $g_\ell : [-1, 1] \to [-1, 1]$.

**Corollary 4.** *Let $S \subset S^{d-1}$ with $|S| = n \geq 2$. Let $r := \max_{x,y\in S, x\neq y}\frac{2}{\sqrt{1-|\langle x,y\rangle|}}$. Let $\ell \in \mathbb{N}^+$.*

*Let $\delta > 0$ be such that $\delta < \frac{2}{r^2}$. Then there exists a random map $\varphi_\ell : \mathbb{R}^d \to D_\ell^{-1/2}\{-1, 1\}^{D_\ell}$ (independent of $S$ except through parameters $n, d, r$) such that with probability $(1 - \frac{1}{n})^\ell$ it satisfies*

$$\left|\langle\varphi_\ell(x), \varphi_\ell(y)\rangle - f_\ell(\langle x, y\rangle)\right| < \delta$$

*for all $x, y \in S$, where $D_\ell \leq \lceil\frac{24\log n}{\delta^2}\rceil$.*

The above corollary follows immediately from the following theorem which also determines a bound on the error of the approximation accuracy not only of $f_\ell(\langle x, y\rangle)$ by $\langle\varphi_\ell(x), \varphi_\ell(y)\rangle$, but also the approximation accuracy of $f_j(\langle x, y\rangle)$ by $\langle\varphi_j(x), \varphi_j(y)\rangle$ for all $j \in \{1, \ldots, \ell\}$.

**Theorem 5.** *Let $S \subset S^{d-1}$ with $|S| = n \geq 2$. Let $r := \max_{x,y\in S, x\neq y}\frac{2}{\sqrt{1-|\langle x,y\rangle|}}$. Let $\ell \in \mathbb{N}^+$.*

*Let $\delta > 0$ be such that $\delta < \frac{2}{r^2}$. Then there exist random maps $\varphi_j : \mathbb{R}^d \to D_j^{-1/2}\{-1, 1\}^{D_j}$, $j \in \{1, \ldots, \ell\}$ (independent of $S$ except through parameters $n, d, r$) such that for all $j \in \{1, \ldots, \ell\}$, with probability at least $(1 - \frac{2}{n})^j$, $\varphi_j$ satisfies*

$$\left|\langle\varphi_j(x), \varphi_j(y)\rangle - f_j(\langle x, y\rangle)\right| < \frac{\delta}{2^{\ell-j}r^{3((2/3)^j-(2/3)^\ell)}} \tag{2}$$

*for all $x, y \in S$, where $D_j \leq \left\lceil\frac{24\cdot2^{2(\ell-j)}r^{6((2/3)^j-(2/3)^\ell)}\log n}{\delta^2}\right\rceil$.*

*Proof.* The maps $\varphi_j$ will be compositions of maps of the following form. Set $D \in \mathbb{N}^+$. Let $Z_i$, $1 \leq i \leq D$ be i.i.d. standard Gaussian random vectors in $\mathbb{R}^d$. Let $\varphi^D : \mathbb{R}^d \to \mathbb{R}^D$ be defined by

$$\varphi^D(x)_i := D^{-1/2}\operatorname{sign}(\langle x, Z_i\rangle),$$

where $\varphi^D(x)_i$ is the $i$th coordinate of $\varphi^D(x)$. A direct calculation, see [26], shows that $\mathbb{E}\langle\varphi^D(x), \varphi^D(y)\rangle = f(\langle x, y\rangle)$. Furthermore, $\|\varphi^D(x)\| = 1$ for all $x \in \mathbb{R}^d$.

First we use this construction to define $\varphi_1$. We let $\varphi_1 = \varphi^{D_1}$ where $D_1$ is chosen below. Using Hoeffding's inequality, for all $x, y \in S$,

$$\mathbb{P}\left(\left|\langle\varphi(x), \varphi(y)\rangle - f(\langle x, y\rangle)\right| > \frac{\delta}{2^{\ell-1}r^{3((2/3)-(2/3)^\ell)}}\right)$$

$$= \mathbb{P}\left(\left|D_1\langle\varphi(x), \varphi(y)\rangle - D_1 f(\langle x, y\rangle)\right| > \frac{D_1\delta}{2^{\ell-1}r^{3((2/3)-(2/3)^\ell)}}\right)$$

$$\leq 2\exp\left(-\frac{D_1\delta^2}{2\cdot2^{2(\ell-1)}r^{6((2/3)-(2/3)^\ell)}}\right).$$

We set $D_1 = \left\lceil\frac{6\cdot2^{2(\ell-1)}r^{6((2/3)-(2/3)^\ell)}\log n}{\delta^2}\right\rceil$. This means that the above probability is less than $2/n^3$ and that $\varphi_1$ satisfies the conditions of the theorem with probability at least $1 - \binom{n}{2}\frac{2}{n^3} \geq 1 - 1/n \geq 1 - 2/n$.

Now assume that the required map exists for some $j \geq 1$. We will show that it exists for $j + 1$. So there exists a map $\varphi_j : \mathbb{R}^d \to D_j^{-1/2}\{-1, 1\}^{D_j}$ which satisfies Eq. (2) with probability at least

$(1 - \frac{2}{n})^j$. Let $\varphi_{j+1}$ be defined by $\varphi_{j+1}(x) = \varphi^{D_{j+1}}(\varphi_j(x))$ where $D_{j+1}$ will be chosen at the end of the proof. We will show that, conditioned on the event that $\varphi_j$ does satisfy Eq. (2), the probability that $\varphi_{j+1}$ satisfies Eq. (2) is at least $1 - \frac{2}{n}$. So assume that $\varphi_j$ does satisfy Eq. (2). Recall that we want to show that $\langle \varphi_{j+1}(x), \varphi_{j+1}(y) \rangle$ is a good estimate of $f_{j+1}(\langle x, y \rangle)$. We have by [26] that

$$\mathbb{E}\langle \varphi_{j+1}(x), \varphi_{j+1}(y) \rangle = f(\langle \varphi_j(x), \varphi_j(y) \rangle),$$

i.e., $\langle \varphi_{j+1}(x), \varphi_{j+1}(y) \rangle$ is an unbiased estimator of $f(\langle \varphi_j(x), \varphi_j(y) \rangle)$. By the triangle inequality,

$$\left| \langle \varphi_{j+1}(x), \varphi_{j+1}(y) \rangle - f_{j+1}(\langle x, y \rangle) \right|$$

$$\leq \left| \langle \varphi_{j+1}(x), \varphi_{j+1}(y) \rangle - f(\langle \varphi_j(x), \varphi_j(y) \rangle) \right| + \left| f(\langle \varphi_j(x), \varphi_j(y) \rangle) - f(f_j(\langle x, y \rangle)) \right|. \quad (3)$$

First we give a bound on the second term in Eq. (3). We are assuming that $\varphi_j$ satisfies Eq. (2), i.e., that for all $x, y \in S$,

$$\left| \langle \varphi_j(x), \varphi_j(y) \rangle - f_j(\langle x, y \rangle) \right| \leq \frac{\delta}{2^{\ell-j} r^{3((2/3)^j - (2/3)^\ell)}}.$$

So to get a bound on the second term in Eq. (3) we need to get an upper bound on the derivative of $f$ in the interval between $\langle \varphi_j(x), \varphi_j(y) \rangle$ and $f_j(\langle x, y \rangle)$. We claim for all pairs $x, y \in S$, $x \neq y$, the derivative of $f$ in the interval between $\langle \varphi_j(x), \varphi_j(y) \rangle$ and $f_j(\langle x, y \rangle)$ is upper bounded by $r^{(2/3)^j}$.

Let $t$ be in the closed interval between $\langle \varphi_j(x), \varphi_j(y) \rangle$ and $f_j(\langle x, y \rangle)$. By definition of $\varphi_j$ and $f_j$ this implies $|t| \leq 1$. We also have $|t| \leq |f_j(\langle x, y \rangle)| + \delta / 2^{\ell-j} r^{3((2/3)^j - (2/3)^\ell)} \leq |f_j(\langle x, y \rangle)| + \delta$ by Eq. (2) and the fact that $r \geq 2$ by definition. Thus,

$$1 - |t| \geq 1 - |f_j(\langle x, y \rangle)| - \delta$$

$$\geq (1 - |\langle x, y \rangle|)^{(2/3)^j} - \delta \qquad \text{by Lemma 10 in Appendix B}$$

$$\geq \frac{(1 - |\langle x, y \rangle|)^{(2/3)^j}}{2} \qquad \text{by } \delta < \frac{2}{r^2} \leq \frac{(1 - |\langle x, y \rangle|)}{2} \leq \frac{(1 - |\langle x, y \rangle|)^{(2/3)^j}}{2}.$$

Using this and $|t| \leq 1$, we get

$$f'(t) = \frac{2}{\pi \sqrt{1 - t^2}} \leq \frac{2}{\pi \sqrt{1 - |t|}} \leq \frac{2\sqrt{2}}{\pi \left( \sqrt{1 - |\langle x, y \rangle|} \right)^{(2/3)^j}} \leq \frac{2\sqrt{2} r^{(2/3)^j}}{\pi} < r^{(2/3)^j}.$$

We have shown that the derivative of $f$ in the interval between $\langle \varphi_j(x), \varphi_j(y) \rangle$ and $f_j(\langle x, y \rangle)$ is upper bounded by $r^{(2/3)^j}$. This combined with Eq. (2) and the fact that the derivative of $f$ is positive implies that

$$\left| f(\langle \varphi_j(x), \varphi_j(y) \rangle) - f(f_j(\langle x, y \rangle)) \right| \leq r^{(2/3)^j} \frac{\delta}{2^{\ell-j} r^{3((2/3)^j - (2/3)^\ell)}}$$

$$= \frac{\delta}{2 \cdot 2^{\ell-(j+1)} r^{3((2/3)^{j+1} - (2/3)^\ell)}},$$

where the equality above uses that $3((2/3)^j - (2/3)^\ell) = \sum_{i=j}^{\ell-1} (2/3)^j$.

Now we deal with the first term in Eq. (3). Using Hoeffding's inequality,

$$\mathbb{P}\left( \left| \langle \varphi_{j+1}(x), \varphi_{j+1}(y) \rangle - f_{j+1}(\langle x, y \rangle) \right| > \frac{\delta}{2 \cdot 2^{\ell-(j+1)} r^{3((2/3)^{j+1} - (2/3)^\ell)}} \right)$$

$$= \mathbb{P}\left( \left| D_{j+1} \langle \varphi_{j+1}(x), \varphi_{j+1}(y) \rangle - D_{j+1} f_{j+1}(\langle x, y \rangle) \right| > \frac{D_{j+1} \delta}{2 \cdot 2^{\ell-(j+1)} r^{3((2/3)^{j+1} - (2/3)^\ell)}} \right)$$

$$\leq 2 \exp\left( -\frac{\delta^2 D_{j+1}}{8 \cdot 2^{2(\ell-(j+1))} r^{6((2/3)^{j+1} - (2/3)^\ell)}} \right).$$

We set $D_{j+1} = \left\lceil \frac{24 \cdot 2^{2(\ell-(j+1))} r^{6((2/3)^{j+1} - (2/3)^\ell)} \log n}{\delta^2} \right\rceil$. This means that the above probability is less than $2/n^3$. So, using Eq. (3) and the previously established bound on the second term in Eq. (3), we have shown that for any pair $x, y \in S$,

$$\left| \langle \varphi_{j+1}(x), \varphi_{j+1}(y) \rangle - f_{j+1}(\langle x, y \rangle) \right| < \frac{\delta}{2^{\ell-(j+1)} r^{3((2/3)^{j+1} - (2/3)^\ell)}}$$

with probability at least $1 - \frac{2}{n^3}$. So, conditioned on the event that $\varphi_j$ satisfies Eq. (2), $\varphi_{j+1}$ satisfies Eq. (2) with probability at least $1 - \frac{2}{n}$. Since the probability that $\varphi_{j+1}$ satisfies Eq. (2) is greater than or equal to the probability that both $\varphi_{j+1}$ and $\varphi_j$ satisfy Eq. (2), this means that the probability that $\varphi_{j+1}$ satisfies Eq. (2) is at least $(1 - \frac{2}{n})^{j+1}$. □

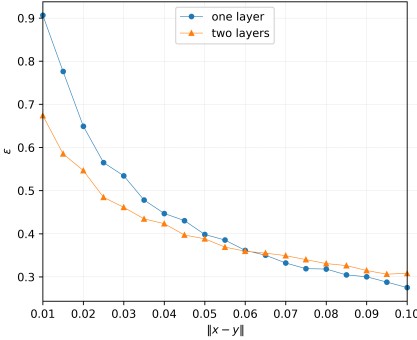 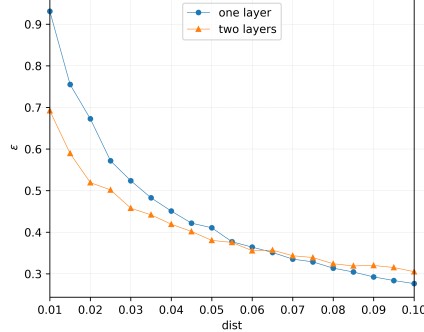

Figure 1: The left, resp. right figure shows the average over 4000 trials of the value of $\epsilon$ such that $2 - 2g_1(\langle\varphi_1(x), \varphi_1(y)\rangle)$ and $2 - 2g_2(\langle\varphi_2(x), \varphi_2(y)\rangle)$ approximate $\|x - y\|^2$ up to a multiplicative $(1 \pm \epsilon)$ error when the input dimension is 2, resp. 2000.

## 3 The recovery of $\|x - y\|^2$ by $\varphi_\ell(x)$ and $\varphi_\ell(y)$

Here we explain how the proofs of the main theorems, Theorems 2 and 3 are completed. This is done in two steps. Recall that we showed in Corollary 4 that $\langle\varphi_\ell(x), \varphi_\ell(y)\rangle$ is a good approximation of $f_\ell(\langle x, y\rangle)$ for all pairs $x, y \in S$. The first step is Theorem 6 which shows that $g_\ell(\langle\varphi_\ell(x), \varphi_\ell(y)\rangle)$ is a good approximation of $\langle x, y\rangle$. The reason this is true is because $g_\ell$ is the inverse of $f_\ell$. So the proof of Theorem 6 uses facts about the derivative of $g_\ell$ (in particular Theorem 7 in Appendix B) to show that the bound on the error of the approximation of $f_\ell(\langle x, y\rangle)$ by $\langle\varphi_\ell(x), \varphi_\ell(y)\rangle$ implies a bound on the error of the approximation of $\langle x, y\rangle$ by $g_\ell(\langle\varphi_\ell(x), \varphi_\ell(y)\rangle)$ (Theorem 6).

The second step is to set $\ell = \lceil\log_2\log_2\frac{4}{m}\rceil$ where $m$ is the minimum distance between pairs of distinct points in $S$ and then show using the polarization identity that the error bound established in Theorem 6 implies the error bounds in Theorems 2 and 3. The proof of the following theorem and the proof that Theorems 2 and 3 follow from Theorem 6 are in Appendix A.

**Theorem 6.** *Let $S \subset S^{d-1}$ with $|S| = n$. Let $\ell \in \mathbb{N}^+$ and $\epsilon > 0$ and assume that $\epsilon$ satisfies $\epsilon < 1 - |\langle x, y\rangle|$ for all $x, y \in S$ with $x \neq y$. Then the random map $\varphi_\ell : S^{d-1} \to N^{-1/2}\{-1, 1\}^N$ from Theorem 5 satisfies that, with probability at least $(1 - 2/n)^\ell$, for all $x, y \in S$, $g_\ell(\langle\varphi_\ell(x), \varphi_\ell(y)\rangle)$ is equal to $\langle x, y\rangle$ up to an additive $\pm\epsilon\|x - y\|^{2 - 2^{-\ell+1}}$ error where $N = \left\lceil\frac{48(\pi/\sqrt{2})^{2\ell}\log n}{\epsilon^2}\right\rceil$. Equivalently, for all $x, y \in S$, $2 - 2g_\ell(\langle\varphi_\ell(x), \varphi_\ell(y)\rangle)$ is equal to $\|x - y\|^2$ up to an additive $\pm\epsilon\|x - y\|^{2 - 2^{-\ell+1}}$ error.*

## 4 Experiments

The main goal of our experiments is to validate the idea of composing multiple random feature maps. Our main results show that better error guarantees can be obtained by composing multiple random feature maps. This is demonstrated by the additive error term $\pm\epsilon\|x - y\|^{2 - 2^{-\ell+1}}$ in Theorem 6 which is smaller for larger values of $\ell$ as long as $\|x - y\| < 1$. Recall that we refer to the parameter $\ell$ as the number of *layers* of the random mapping. In our experiments, we will focus on comparing the one layer maps $\varphi_1$ to the two layer maps $\varphi_2$. First we do a simple experiment to determine more specifically under what conditions we should expect two layers to outperform one. Then we apply the maps to nearest neighbor search and again compare the performance of one layer to two layers.

### 4.1 Two layers vs. one layer

As already mentioned, Theorem 6 indicates that the additive error obtained by the one layer map is proportional to $\|x - y\|$ and the additive error obtained by the two layer map is proportional to $\|x - y\|^{3/2}$. Therefore, two layers should give better error estimates than one layer when $\|x - y\|$ is sufficiently small. Our first experiment answers the following question; how small must $\|x - y\|$

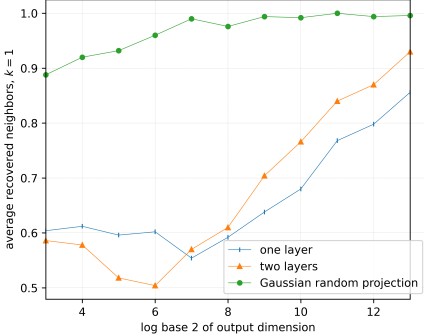 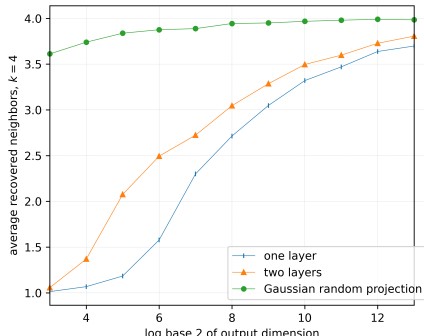

Figure 2: Average over 500 trials of the number of true $k$ nearest neighbors of $X_0$ in $D$ recovered from $\varphi_1(D)$, $\varphi_2(D)$, and $GD$ where $G$ is an i.i.d. Gaussian random matrix.

be so that two layers outperform one? We consider pairs of points $x, y \in S^{d-1}$ where $\|x - y\|$ ranges from .01 to .1. We will see in this experiment that the input dimension has little to no effect and so we can without loss of generality assume $d = 2$. In the $d = 2$ case we take $x = (1, 0)$ and $y = (a, \sqrt{1 - a^2})$ where $a$ is chosen so that $\|x - y\|$ ranges from .01 to .1 in increments of .005, see Fig. 1. We map $x, y$ by both maps $\varphi_1$ and $\varphi_2$ where the output dimension of both maps is 1000 and $\varphi_2$ first maps to $\{-1, 1\}^{6000}$. In Fig. 1 we plot the average over 4000 trials of the value of $\epsilon$ such that $2 - 2g_1(\langle \varphi_1(x), \varphi_1(y) \rangle)$ and $2 - 2g_2(\langle \varphi_2(x), \varphi_2(y) \rangle)$ approximate $\|x - y\|^2$ up to a multiplicative $(1 \pm \epsilon)$ error. We see in Fig. 1 that the two layer map gives a better approximation of $\|x - y\|^2$ when $\|x - y\| \leq .06$ and the one layer map gives a better approximation of $\|x - y\|^2$ when $\|x - y\| > .06$. From a practical perspective, this means that the one layer map may often outperform the two layer map because most real world datasets have few points at this small of a distance. However, very large datasets may be more likely to have distances in the range where two layers is better. We do this same experiment when $d = 2000$ except for $d = 2000$ we choose the input points randomly: We let $x$ be uniform on $S^{d-1}$ and set $y = z/\|z\|$ where $z = x + (\text{dist}/\sqrt{d})N(0, I_d)$ and dist ranges from .01 to .1. With this choice, $\mathbb{E}\|x - y\| \approx \text{dist}$. Because there is no dependence on the input dimension in our error bounds, we expect the error to depend on $\|x - y\|$ but not $d$, which is verified in Fig. 1.

## 4.2 Two layers vs. one layer; nearest neighbor search

Given a set of points and a query point, nearest neighbor search is the task of finding the points that are closest to the given query point. Nearest neighbor search is used in many applications to solve regression/classification problems. Before performing nearest neighbor search, some dimensionality reduction or compression method can be used to reduce the computational cost. We test the performance of our maps $\varphi_\ell$ for this task.

Again we focus on $\ell = 1, 2$. Given a data set $D$ and a query point $X \in D$, we first map $D$ by $\varphi_1$ and $\varphi_2$ to $N^{-1/2}\{-1, 1\}^N$. We then calculate the $k$ nearest neighbors of $X$ according to the compressed data and compare how many of the true $k$ nearest neighbors of $X$ in $D$ are recovered.

*Randomly generated data.* We let $X_0$ (the query point) be a uniform random vector on the unit sphere $S^2$. Then for $i \in \{1, 2, \ldots, 100\}$, we let $Y_i = X_0 + (4/5)(i/200)N(0, I_3)$ where $N(0, I_3) \subset \mathbb{R}^3$ is from the multivariate normal distribution (and the $Y_i$ are independent). The constants are chosen so that $\mathbb{E}\|X_0 - Y_i\| \approx i/200$. Then the data $D$ is defined to be $X_0$ along with $\{Y_i/\|Y_i\|\}_{i \in \{1, 100\}}$. We map $D$ by $\varphi_1$ and $\varphi_2$ where the output dimension ranges from $2^3$ to $2^{13}$ and $\varphi_2$ first maps to the space of dimension six times the output dimension. We also map $D$ by a standard Gaussian random matrix to $\mathbb{R}^d$ with the same range of output dimensions. The average number of $k$ nearest neighbors recovered is shown in Fig. 2. As is to be expected based on the previous experiment, two layers generally outperform one. We remark that the accuracy of our maps is not meant to be compared to the accuracy of the Gaussian random matrix because that mapping uses full precision real numbers and does not convert to bits.

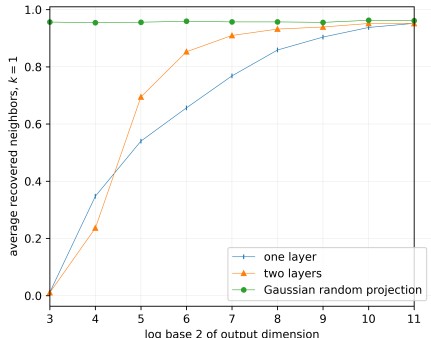 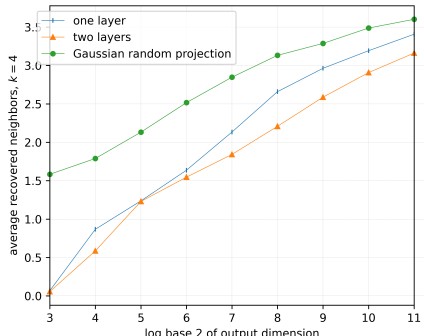

Figure 3: Average over 10 independent trials and over all 179 query points of the number of true $k$ nearest neighbors of 179 query points in RCV1 recovered from $\varphi_1(D)$, $\varphi_2(D)$, and $GD$ where $G$ is an i.i.d. Gaussian random matrix.

*Real data.* We perform a similar experiment with the RCV1 dataset [20]. The RCV1 data set consists of 804414 samples and 47236 features. We only consider the first 23149 samples which have been previously designated as the training set. As query points, we select all data points which have at least one neighbor at distance less than .05. There are 179 such points. The RCV1 dataset consists of unit norm vectors. This is our data set $D$. We map $D$ by $\varphi_1$ and $\varphi_2$ where the output dimension ranges from $2^3$ to $2^{11}$ and $\varphi_2$ first maps to the space of dimension six times the output dimension. We also map $D$ by a standard Gaussian random matrix to $\mathbb{R}^d$ with the same range of output dimensions. The average number of $k$ nearest neighbors recovered is shown in Fig. 3. We see that in the case $k = 1$, the two layer map outperforms the one layer map when the output dimension is sufficiently large. This again confirms expectations based on Fig. 1. In the $k = 4$ case, the one layer map is superior. This is likely due to the fact that while our query points all have at least one neighbor at distance at most .05, the other three out of the four nearest neighbors may be at a significantly greater distance. Therefore, again considering Fig. 1, it is not surprising that the one layer map gives a better approximation of the four nearest neighbors.

## 5   Conclusion

We introduced a new method for compressing point sets while maintaining information about approximate distances between pairs of points. The method compresses point sets using a composition $\varphi_\ell$ of $\ell$ random feature mappings. The main advantage of composing multiple feature maps is that, rather than approximating pairwise distances up to an *additive* $\epsilon$ error, our maps accomplish the more difficult task of approximating the distances up to a *multiplicative* $(1 \pm \epsilon)$ error. The reason that we get multiplicative rather than additive error guarantees is a direct result of composing multiple random feature maps and has to do with the behavior of the derivative of the function $g_\ell$ (introduced in Section 1.1) near $t = 1$. We also validate the idea of composing multiple random feature maps experimentally.

## Acknowledgments and Disclosure of Funding

This material is based upon work supported by the National Science Foundation under Grants CCF-1934568 and CCF-2006994.

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

# A   Proofs of Theorems 2, 3, and 6

*Proof of Theorem 6.* We will actually prove the stronger result that with probability at least $(1 - 2/n)^\ell$, for all $x, y \in S$, $g_\ell(\langle \varphi_\ell(x), \varphi_\ell(y) \rangle)$ is equal to $\langle x, y \rangle$ up to an additive $\pm \epsilon(2 - 2|\langle x, y \rangle|)^{1 - 2^{-\ell}}$ error. This statement implies the theorem statement by the polarization identity.

Set $\delta = (\epsilon/\sqrt{2})(\sqrt{2}/\pi)^\ell$ in Theorem 5. One can check that the assumption that $\epsilon < 1 - |\langle x, y \rangle|$ for all $x, y \in S$ with $x \neq y$ implies that $\delta < 2/r^2$ for all $x, y \in S$ with $x \neq y$ as required by Theorem 5. From Theorem 5, the map $\varphi_\ell : \mathbb{R}^d \to \frac{1}{\sqrt{N}}\{-1, 1\}^N$ satisfies, with probability at least $(1 - 2/n)^\ell$, that

$$\left| \langle \varphi_\ell(x), \varphi_\ell(y) \rangle - f_\ell(\langle x, y \rangle) \right| < (\epsilon/\sqrt{2})(\sqrt{2}/\pi)^\ell \tag{4}$$

for all $x, y \in S$ where $N = \left\lceil \frac{48(\pi/\sqrt{2})^{2\ell} \log n}{\epsilon^2} \right\rceil$. Now assume that $\varphi_\ell$ does satisfy Eq. (4) for all $x, y \in S$. Since $g_\ell\big(f_\ell(t)\big) = t$ for all $t \in [-1, 1]$, Eq. (4) implies that $g_\ell(\langle \varphi_\ell(x), \varphi_\ell(y) \rangle)$ should be a good approximation of $\langle x, y \rangle$. In particular, we claim that for all $x, y \in S$, $g_\ell(\langle \varphi_\ell(x), \varphi_\ell(y) \rangle)$ is equal to $\langle x, y \rangle$ up to an additive error of $\pm \epsilon(2 - 2|\langle x, y \rangle|)^{1 - 2^{-\ell}}$. In order to show this we first need to get a bound on the derivative of $g_\ell$ in the interval between $\langle \varphi_\ell(x), \varphi_\ell(y) \rangle$ and $f_\ell(\langle x, y \rangle)$.

Let $t$ be in the interval between $\langle \varphi_\ell(x), \varphi_\ell(y) \rangle$ and $f_\ell(\langle x, y \rangle)$. By Theorem 7,

$$g_\ell'(t) \leq \frac{\pi^\ell}{2^{\frac{\ell+1}{2}}} \big(2 - 2g_\ell(|t|)\big)^{1 - 2^{-\ell}}.$$

By Eq. (4), we have that $\big||f_\ell(\langle x, y \rangle)| - |t|\big| \leq \sqrt{2}\epsilon(\frac{\sqrt{2}}{\pi})^\ell$. By Lemma 8 this means that $g_\ell\big(|f_\ell(\langle x, y \rangle)|\big) - g_\ell(|t|) \leq \sqrt{2}\epsilon(\sqrt{2}/\pi)^\ell(\pi/2)^\ell \leq \epsilon$, i.e. that $g_\ell(|t|) \geq g_\ell\big(|f_\ell(\langle x, y \rangle)|\big) - \epsilon$. Since $f$ is an odd function, $f_\ell(t)$ is also an odd function and so $g_\ell\big(|f_\ell(\langle x, y \rangle)|\big) = g_\ell\big(f_\ell(|\langle x, y \rangle|)\big) = |\langle x, y \rangle|$. So we have shown that $g_\ell(|t|) \geq |\langle x, y \rangle| - \epsilon$. Since $\epsilon < 1 - |\langle x, y \rangle|$, we have that $2 - 2g_\ell(|t|) \leq 2 - 2|\langle x, y \rangle| + 2\epsilon < 2(2 - 2|\langle x, y \rangle|)$. Using that $2^{1 - 2^{-\ell}} \leq 2$, this means that

$$g_\ell'(t) \leq \frac{\pi^\ell}{2^{\frac{\ell+1}{2}}} \big(2 - 2g_\ell(|t|)\big)^{1 - 2^{-\ell}} \leq \sqrt{2} \left(\frac{\pi}{\sqrt{2}}\right)^\ell (2 - 2|\langle x, y \rangle|)^{1 - 2^{-\ell}}.$$

This bound on the derivative along with the fact that $g_\ell'(t) > 0$ and Eq. (4) means that

$$\left| g_\ell\big(\langle \varphi_\ell(x), \varphi_\ell(y) \rangle\big) - g_\ell\big(f_\ell(\langle x, y \rangle)\big) \right| < \epsilon\big(2 - 2|\langle x, y \rangle|\big)^{1 - 2^{-\ell}}.$$

Since $g_\ell\big(f_\ell(\langle x, y \rangle)\big) = \langle x, y \rangle$, the theorem follows. $\qquad\square$

Now that we have established the above result, we can prove Theorems 2 and 3. We restate the theorems from the intro for the sake of readability.

**Theorem 2.** *Let* $S \subset S^{d-1}$ *with* $|S| = n \geq 2$. *Let* $m = \min_{x,y \in S, x \neq y} \|x - y\|$ *and* $\ell = \lceil \log_2 \log_2 \frac{4}{m} \rceil \geq 1$. *Let* $\epsilon > 0$ *and assume that* $\epsilon < \min_{x,y \in S, x \neq y} 1 - |\langle x, y \rangle|$. *Then the random map* $\varphi_\ell : S^{d-1} \to \frac{1}{\sqrt{N}}\{-1, 1\}^N$ *with* $N = \Theta\big(\frac{\log n}{\epsilon^2}(\log \frac{1}{m})^{2 \log_2(\pi/\sqrt{2})}\big)$ *(defined in the proof of Theorem 5 and independent of $S$ except through parameters $n, d, m$ and $\min_{x,y \in S, x \neq y} 1 - |\langle x, y \rangle|$) satisfies the following with probability at least* $(1 - \frac{2}{n})^\ell$: *$\varphi_\ell(S)$ is a sketch of $S$ that allows one to recover all squared distances between pairs of points in $S$ up to a multiplicative $(1 \pm \epsilon)$ error. The number of bits of the sketch is $\Theta\big(\frac{n \log n}{\epsilon^2}(\log \frac{1}{m})^{2 \log_2(\pi/\sqrt{2})}\big)$.*

*Proof.* Let $\varphi_\ell : S^{d-1} \to N^{-1/2}\{-1, 1\}^N$ be the map from Theorem 5 that by Theorem 6 satisfies, with probability at least $(1 - 2/n)^\ell$, that $2 - 2g_\ell(\langle \varphi_\ell(x), \varphi_\ell(y) \rangle)$ is equal to $\|x - y\|^2$ up to an additive $\pm(\epsilon/4)\|x - y\|^{2 - 2^{-\ell+1}}$ error for all $x, y \in S$ with $N = \left\lceil \frac{768(\pi/\sqrt{2})^{2\ell} \log n}{\epsilon^2} \right\rceil$. Now assume that $\varphi_\ell$ does satisfy this condition for all $x, y \in S$. We have

$$\|x - y\|^{-2^{-\ell+1}} \leq (1/m)^{2^{-\ell+1}} = \big((1/m)^{2^{-\ell}}\big)^2 \leq \big((1/m)^{\frac{1}{\log_2(4/m)}}\big)^2 < 4.$$

This means that we actually estimate $\|x-y\|^2$ up to an additive $\pm\epsilon\|x-y\|^2$ error, i.e., a multiplicative $(1\pm\epsilon)$ error. We have

$$\left(\frac{\pi}{\sqrt{2}}\right)^{2\ell} \leq \left(\frac{\pi}{\sqrt{2}}\right)^{2(1+\log_2\log_2\frac{4}{m})} = \left(\frac{\pi^2}{2}\right)2^{2\log_2(\frac{\pi}{\sqrt{2}})\log_2\log_2(\frac{4}{m})} = \frac{\pi^2}{2}\left(\log_2\frac{4}{m}\right)^{2\log_2\frac{\pi}{\sqrt{2}}}$$

so the result follows. $\qquad\square$

The above theorem shows that given a set of points $S \subset \mathbb{R}^d$, there exists an appropriate choice of $\ell$ and $N$ so that the random map $\varphi_\ell : S^{d-1} \to N^{-1/2}\{-1,1\}^N$ satisfies, with high probability, that $\varphi_\ell(S)$ is a sketch of $S$ that allows one to recover all squared distances between points in $S$ up to a multiplicative $(1\pm\epsilon)$ error. The next theorem shows that this same sketching algorithm also works for point sets that do not necessarily consist of unit norm points provided that the sketch also stores the approximate norms of points in $S$.

**Theorem 3.** *Let $S \subset B^d$ with $|S| = n \geq 2$ and set $\rho = \min_{x\in S}\|x\|^2$. Let $m = \min_{x,y\in S,x\neq y}\|\hat{x}-\hat{y}\|$ and $\ell = \lceil\log_2\log_2\frac{4}{m}\rceil \geq 1$. Let $\epsilon > 0$ and assume that $\epsilon < \min_{x,y\in S,x\neq y} 1 - |\langle\hat{x},\hat{y}\rangle|$. Then the random map $\varphi_\ell : S^{d-1} \to \frac{1}{\sqrt{N}}\{-1,1\}^N$ with $N = \Theta\left(\frac{\log n}{\epsilon^2}(\log\frac{1}{m})^{2\log_2(\pi/\sqrt{2})}\right)$ (defined in the proof of Theorem 5 and independent of $S$ except through parameters $n, d, m$ and $\min_{x,y\in S,x\neq y} 1 - |\langle\hat{x},\hat{y}\rangle|$) satisfies the following with probability at least $(1 - \frac{2}{n})^\ell$: $\varphi_\ell(\hat{S})$ and the norm of each point in $S$ up to an additive $\pm\rho m^2\epsilon/48$ error is a sketch of $S$ that allows one to recover all squared distances between pairs of points in $S$ up to a multiplicative $(1\pm\epsilon)$ error. Moreover the number of bits of the sketch is $\Theta\left(\frac{n\log n}{\epsilon^2}\left(\log\frac{1}{m}\right)^{2\log_2(\pi/\sqrt{2})} + n\log\frac{1}{\rho m^2\epsilon}\right)$.*

*Proof.* For each $x \in S$, let $n_x$ be an approximation of $\|x\|$ up to an additive $\pm\rho m^2\epsilon/48$ error.

First we claim that in order to recover squared distances up to a multiplicative $(1\pm\epsilon)$ error, it suffices to recover squared distances to an additive $\pm\epsilon\|x\|\|y\|\|\hat{x}-\hat{y}\|^2$ error. The reason is that for any $x, y \in \mathbb{R}^d$, we can prove the inequality $\epsilon\|x\|\|y\|\|\hat{x}-\hat{y}\|^2 \leq \epsilon\|x-y\|^2$ by observing that $\|x\|\|y\|\|\hat{x}-\hat{y}\|^2 = \|x\|\|y\|(2-2\langle\hat{x},\hat{y}\rangle) = 2\|x\|\|y\| - 2\langle x,y\rangle \leq \|x\|^2 + \|y\|^2 - 2\langle x,y\rangle = \|x-y\|^2$.

Now let $\hat{S} = \{\hat{x} : x \in S\}$. Let $\varphi_\ell : S^{d-1} \to N^{-1/2}\{-1,1\}^N$ with $N = \Theta\left(\frac{(\pi/\sqrt{2})^{2\ell}\log n}{\epsilon^2}\right)$ be the map from Theorem 5 that by Theorem 6 satisfies, with probability at least $(1-2/n)^\ell$, that $g_\ell(\langle\varphi_\ell(\hat{x}),\varphi_\ell(\hat{y})\rangle)$ is equal to $\langle\hat{x},\hat{y}\rangle$ up to an additive $\pm(\epsilon/32)(2 - 2|\langle\hat{x},\hat{y}\rangle|)^{1-2^{-\ell}}$ error for all $\hat{x},\hat{y} \in \hat{S}$. Assume that $\varphi_\ell$ does satisfy this condition for all $\hat{x},\hat{y} \in \hat{S}$. Notice that this implies that $g_\ell(\langle\varphi_\ell(\hat{x}),\varphi_\ell(\hat{y})\rangle)$ is equal to $\langle\hat{x},\hat{y}\rangle$ up to an additive $\pm(\epsilon/32)(2 - 2\langle\hat{x},\hat{y}\rangle)^{1-2^{-\ell}}$ error for all $\hat{x},\hat{y} \in \hat{S}$. We have

$$(2 - 2\langle\hat{x},\hat{y}\rangle)^{-2^{-\ell}} \leq (1/m)^{2^{-\ell+1}} = \left((1/m)^{2^{-\ell}}\right)^2 \leq \left((1/m)^{\frac{1}{\log_2(4/m)}}\right)^2 < 4.$$

This means that $g_\ell(\langle\varphi_\ell(\hat{x}),\varphi_\ell(\hat{y})\rangle)$ is equal to $\langle\hat{x},\hat{y}\rangle$ up to an additive $\pm(\epsilon/8)(2 - 2\langle\hat{x},\hat{y}\rangle)$ error for all $\hat{x},\hat{y} \in \hat{S}$, i.e., an additive $\pm(\epsilon/8)\|\hat{x}-\hat{y}\|^2$ error. For any $x, y \in S$,

$$n_xn_yg_\ell(\langle\varphi_\ell(\hat{x}),\varphi_\ell(\hat{y})\rangle) \leq \left(\|x\| + \rho m^2\epsilon/48\right)\left(\|y\| + \rho m^2\epsilon/48\right)\left(\langle\hat{x},\hat{y}\rangle + (\epsilon/8)\|\hat{x}-\hat{y}\|^2\right)$$
$$\leq \langle x,y\rangle + (\epsilon/4)\|x\|\|y\|\|\hat{x}-\hat{y}\|^2$$

where the second inequality uses the definition of $\rho$ and $m$. We can also show that

$$n_xn_yg_\ell(\langle\varphi_\ell(\hat{x}),\varphi_\ell(\hat{y})\rangle) \geq \langle x,y\rangle - (\epsilon/4)\|x\|\|y\|\|\hat{x}-\hat{y}\|^2.$$

This means that $n_xn_yg_\ell(\langle\varphi_\ell(\hat{x}),\varphi_\ell(\hat{y})\rangle)$ approximates $\langle x,y\rangle$ up to an additive $\pm(\epsilon/4)\|x\|\|y\|\|\hat{x}-\hat{y}\|^2$ error. Since $n_x$ approximates $\|x\|$ up to an additive $\pm(\epsilon/24)\min_{x,y\in S}\|x\|\|y\|\|\hat{x}-\hat{y}\|^2$ error this means that $n_x^2$ approximates $\|x\|^2$ up to at least an additive $\pm(\epsilon/4)\|x\|\|y\|\|\hat{x}-\hat{y}\|^2$ error. Now this means that

$$n_x^2 + n_y^2 - 2n_xn_yg_\ell(\langle\varphi_\ell(\hat{x}),\varphi_\ell(\hat{y})\rangle)$$

approximates

$$\|x\|^2 + \|y\|^2 - 2\langle x,y\rangle = \|x-y\|^2$$

up to an additive $\pm\epsilon\|x\|\|y\|\|\hat{x}-\hat{y}\|^2$ error.

Storing all the norms of the points up to an additive $\pm\rho$ error requires $\log(1/\rho)$ bits per point.

We have
$$(\pi/\sqrt{2})^{2\ell} \leq (\pi/\sqrt{2})^{2(1+\log_2\log_2 r)} = (\pi^2/2)2^{2\log_2(\pi/\sqrt{2})\log_2\log_2 r} = (\pi^2/2)(\log_2 r)^{2\log_2(\pi/\sqrt{2})}$$
so the result follows. $\qquad\square$

## B    Technical lemmas

**Theorem 7.** *For all $t \in [0,1]$,*
$$g'_\ell\big(f_\ell(t)\big) \leq \frac{\pi^\ell}{2^{\frac{\ell+1}{2}}}(2-2t)^{1-2^{-\ell}}.$$

*This implies that for all $t \in [0,1]$,*
$$g'_\ell(t) \leq \frac{\pi^\ell}{2^{\frac{\ell+1}{2}}}\big(2-2g_\ell(t)\big)^{1-2^{-\ell}}.$$

*and that for all $t \in [-1,1]$,*
$$g'_\ell(t) \leq \frac{\pi^\ell}{2^{\frac{\ell+1}{2}}}\big(2-2g_\ell(|t|)\big)^{1-2^{-\ell}}.$$

*Proof.* We start by proving the first claim.

We will use the inequality
$$1 - f(t) = \frac{2}{\pi}\arccos(t) \leq \sqrt{1-t} \text{ for } t \in [0,1], \tag{5}$$

which follows by finding critical points of $\sqrt{1-t} - \frac{2}{\pi}\arccos(t)$. Now this enables us to prove by induction that
$$1 - f_\ell(t) \leq (2-2t)^{2^{-\ell}} \text{ for } t \in [0,1]$$
for all $\ell \in \mathbb{N}^+$. The base case $\ell = 1$ is (using Eq. (5))
$$1 - f(t) = \frac{2}{\pi}\arccos(t) \leq \sqrt{1-t} \leq \sqrt{2-2t}.$$

The induction step again uses Eq. (5) and also uses that $f_{\ell-1}(t) \in [0,1]$ if $t \in [0,1]$. For all $t \in [0,1]$, we have
$$\begin{aligned}
1 - f_\ell(t) &= 1 - f(f_{\ell-1}(t)) \\
&\leq \sqrt{1 - f_{\ell-1}(t)} \\
&\leq \sqrt{(2-2t)^{2^{-\ell+1}}} \\
&= (2-2t)^{2^{-\ell}}.
\end{aligned}$$

So $1 - f_\ell(t) \leq (2-2t)^{2^{-\ell}}$ for $t \in [0,1]$ follows.

Now we will prove the first theorem claim by induction. The base case is
$$g'\big(f(t)\big) = \frac{\pi}{2}\cos\big(\arcsin(t)\big) = \frac{\pi}{2}\sqrt{1-t^2} \leq \frac{\pi}{2}\sqrt{2-2t}.$$

Now for the induction step, assume that the first theorem claim holds for $f_{\ell-1}$. We first need to establish that for all $t \in [-1,1]$,
$$\begin{aligned}
g'\big(f_\ell(t)\big) &= \frac{\pi}{2}\cos\big((\pi/2)f_\ell(t)\big) \\
&= \frac{\pi}{2}\cos\big(\arcsin(f_{\ell-1}(t))\big) \\
&= \frac{\pi}{2}\sqrt{1-(f_{\ell-1}(t))^2} \\
&\leq \frac{\pi}{\sqrt{2}}\sqrt{1-f_{\ell-1}(t)} &&\text{by } f_{\ell-1}(t) \leq 1 \\
&\leq \frac{\pi}{\sqrt{2}}(2-2t)^{2^{-\ell}}.
\end{aligned}$$

Using the chain rule, $g'_\ell(t) = g'(t)g'_{\ell-1}\big(g(t)\big)$. So for all $t \in [0, 1]$,

$$g'_\ell\big(f_\ell(t)\big) = g'\big(f_\ell(t)\big)g'_{\ell-1}\big(g(f_\ell(t))\big)$$
$$= g'\big(f_\ell(t)\big)g'_{\ell-1}\big(f_{\ell-1}(t)\big)$$
$$\leq \frac{\pi}{\sqrt{2}}(2 - 2t)^{2^{-\ell}}\frac{\pi^{\ell-1}}{2^{\frac{\ell}{2}}}(2 - 2t)^{1 - 2^{-\ell+1}}$$
$$= \frac{\pi^\ell}{2^{\frac{\ell+1}{2}}}(2 - 2t)^{1 - 2^{-\ell}}.$$

The second claim follows by plugging in $g_\ell(t)$ into the first claim. The third claim follows by observing that if $t \in [-1, 0]$, then $g'_\ell(t) = g'_\ell(|t|)$ since $g'_\ell$ is an even function (by the fact that $g_\ell$ is odd.) $\qquad\square$

**Lemma 8.** *For all $t \in [-1, 1]$, $0 \leq g'_\ell(t) \leq (\pi/2)^\ell$.*

*Proof.* By induction. When $\ell = 1$, $g'(t) = \frac{\pi}{2}\cos(\frac{\pi t}{2}) \in [0, \pi/2]$ when $t \in [-1, 1]$. For the induction step, we need to use the fact that $g_{\ell-1}(t) \in [-1, 1]$ when $t \in [-1, 1]$. Using this,

$$g'_\ell(t) = g'\big(g_{\ell-1}(t)\big)g'_{\ell-1}(t) = \frac{\pi}{2}\cos\left(\frac{\pi}{2}g_{\ell-1}(t)\right)g'_{\ell-1}(t) \in [0, (\pi/2)^\ell]. \qquad\square$$

**Lemma 9.** $|f_\ell(t)| \leq |t|$ *for all $t \in [-1, 1]$ and all $\ell \in \mathbb{N}^+$.*

*Proof.* Since $f_\ell(-t) = -f_\ell(t)$ for all $t \in [-1, 1]$, it suffices to prove that $f_\ell(t) \leq t$ for all $t \in [0, 1]$. The claim follows by induction in $\ell$. $\qquad\square$

**Lemma 10.** *For all $\ell \in \mathbb{N}$ and $t \in [-1, 1]$, $1 - |f_\ell(t)| \geq (1 - |t|)^{(2/3)^\ell}$.*

*Proof.* Since $f_\ell(-t) = -f_\ell(t)$, it suffices to show that $1 - f_\ell(t) \geq (1 - t)^{(2/3)^\ell}$ for all $t \in [0, 1]$.

To prove that $1 - f(t) \geq (1 - t)^{2/3}$ we will use that $\arccos(t) \geq \frac{\pi(1-t)^{1/2}}{2(1+t)^{1/6}}$ for all $t \in [0, 1]$ as shown in [31, Remark 2.1]. Now we have $1 - f(t) = \frac{2}{\pi}\arccos(t) \geq \frac{(1-t)^{1/2}}{(1+t)^{1/6}}$. The result now follows since $\frac{1}{(1+t)^{1/6}} \geq (1 - t)^{1/6}$ for all $t \in (-1, 1]$. This is the base case. The induction step uses the fact that for all $t \in [0, 1]$, $f_\ell(t) \in [0, 1]$. Using this and the base case proven above,

$$1 - f_{\ell+1}(t) \geq \big(1 - f_\ell(t)\big)^{2/3} \geq \left((1 - t)^{(2/3)^\ell}\right)^{2/3} = (1 - t)^{(2/3)^{\ell+1}}. \qquad\square$$

