# OpenReview forum: "Euclidean distance compression via deep random features"
_NeurIPS.cc/2024/Conference — NeurIPS 2024 poster_

### Official Review · Reviewer_f7H3 · 2024-06-14

**Soundness:** 3
**Presentation:** 4
**Contribution:** 3
**Rating:** 6
**Confidence:** 4

**Summary:**

The paper focuses on constructing sketches of point sets via (compositions of) random maps $\varphi_l$ into the discrete cube $N^{-\frac{1}{2}}\{-1,1\}^N$, and describes how to get an estimate of the squared Euclidean distance. The paper explains how the maps $\varphi_l$ are constructed and motivate the choice based on properties of the functions $f, g$ , and has detailed proofs bounding the error of the sketch based on $l$. The paper discusses the limitations of $\varphi_l$, i.e. there is an additive $\epsilon\|x - y\|^{2-2^{1-l}}$ error ; hence for $\|x - y\| > 1$, $\varphi_l, l > 1$ may not be optimal. There are experiments with simulated data to show how $\epsilon$ varies, and how $l = 1, 2$ performs for nearest neighbor search, as well as nearest neighbor search with the RCV1 dataset. Based on the experiments, the paper summarizes conditions in when $l = 1$ and $l = 2$ should be used.

**Strengths:**

- The sketching algorithm outlined here stores data as bits (on the discrete cube $\{-1,1\}^N$) (scaling factor can be applied after storage) ; this saves on space (compared to a sketching algorithm that stores data as doubles or floats)

- Treating $\varphi^D$ as the random map, the idea of applying $\varphi^D$ repeatedly, and finding an appropriate ``inverse" to recover the Euclidean distance is a nifty idea.

- The paper is extremely clear, motivating random projection and sketching, as well as giving detailed proofs and explaining why each step was taking for w.h.p. bounds on the error of the sketch.

To summarize, I feel the strengths of this paper are putting together several ideas: linking the derivative of the function $g \equiv \sin(\pi/2 t)$ to bounding the additive error of the approximation, and carefully explaining what happens if $\varphi^D$ is applied multiple times.

**Weaknesses:**

- The one layer map $\varphi_l$ (lines 111-113), if I am not mistaken, comes from the original sign random projections in [6] (Section 3, Random Hyperplane Based Hash Functions for Vectors). While this was originally used to estimate angles, Li et al (Section 4, Sign Random Projections) looked at estimating $a$ (which is $\langle x,y \rangle$ in the notation of this paper) using $\arccos(..)$, given that the margins are known / data is normalized (which is equivalent to points being on $S^{d-1}$ in the notation of this paper). By the polarization identity, $\|x-y\|$ can be recovered directly from $\langle x,y\rangle$. The experiments run generally show that the one layer map is better than the two layer map, except in cases where $\|x - y\| \leq 0.06$ (line 317), but this seems that the two layer map is only useful in very niche cases. It would be good if there is some reference to sign random projections when referring to the one layer map. Unfortunately, this means any novelty would be for the $l$ layer map, $l \geq 2$.

- The plots in Figure 1 show that $\epsilon$ is generally larger than $\|x - y\|$. Moreover (line 317), the two layer map performs better when the Euclidean distance $\|x-y\| \leq 0.06$, which corresponds to $\theta_{x,y} < 0.06$. I am not exactly sure how realistic a two layer map would be. While it may accurately recover the true nearest neighbor, there may also be false positives, i.e. a point with a farther Euclidean distance but have an estimate that is "closer".

- I found it difficult to replicate the experiment in Figure 3 ; since a quick Google search found the text dataset (not pre-processed into vectors). Moreover, I found the "first project the data with a Gaussian random matrix to $\mathbb R^{5000}$" a bit puzzling (e.g. are the true nearest neighbors the original neighbors, or the true nearest neighbors after projecting with a Gaussian random matrix?)

- I note that computing the second layer map took up substantially more time than just the first layer map (even with $D_1 = 6000, D_2 = 1000$), so I am not sure if the increase in computing time is worth the gain in less error.


References:
Li et al: Improving Random Projections Using Marginal Information, COLT 2006

**Questions:**

- Li et al (Section 4) states "In fact, when $\theta$ is close to $0$ or $\pi$, due to the high nonlinearity, the asymptotic variance formula is not reliable." Can the results of Theorem 6 (in this paper) show that if ``sign random projections" were applied again, an angle $\theta$ close to $0$ would have a lower variance (since a small $\theta$ implies a small Euclidean distance)?

- I am uncomfortable with line 472-473, "Since $g_l(f_l(t)) = t$ ..., $g_l(\langle\varphi_l(x), \varphi_l(y)\rangle)$ should be a good approximation of $\langle x, y\rangle$". I agree that $\mathbb E{\langle \varphi^D(x), \varphi^D(y)\rangle} = f(\langle x,y\rangle)$, and also that $g(\mathbb E{\langle \varphi^D(x), \varphi^D(y)\rangle}) = g(f(\langle x,y\rangle)) = \langle x, y \rangle$. But as the goal is to approximate $\langle x,y\rangle$ (and from it the Euclidean distance), $\mathbb E{g({\langle \varphi^D(x), \varphi^D(y)\rangle})} \neq g(\mathbb E{\langle \varphi^D(x), \varphi^D(y)\rangle})$. Hence I am not sure if the bounds are as accurate (here, I am thinking of Taylor expansions of $\mathbb E{g({\langle \varphi^D(x), \varphi^D(y)\rangle})}$, and bounding remainder terms). I might be wrong, and happy to be corrected.

I have tried $\|x-y\| = 0.03$ with $l = 2$ in a similar experiment in Section 4.1, lines 304-325, and I do see the two layer map outperforming the one layer map, but the three layer map having poor performance compared to the two layer map. However, with a lower $\|x-y\| = 0.0003$, I see that the three layer map outperforms both the two layer map and the one layer map (using the choice of 50000->6000->1000), so the general idea is still right.

- Are the true nearest neighbors the original neighbors, or the true nearest neighbors after projecting with a Gaussian random matrix for the experiments in Fig 3?

- The left plot in Fig1 might be more informative when $\epsilon$s are plotted within the unit circle in order to see comparisons for the one layer and two layer map for some fixed values of output dimension. E.g., plot the corresponding $\epsilon$s for the one layer map / two layer map on the "line" parameterized by $(a, \sqrt{1-a^2})$ as $a$ varies, so a reader can visually see the region where the one layer map performs better, and the two layer map performs better. Admittedly, this is only useful in 2D when it is easy to "convert visually" from angles to Euclidean distances.

- A conclusion (or discussion section) that summarizes the ideas in Section 1 would be good as well - although I think this can be done by placing some parts of Section 1 at the end of the paper.

References:
Li et al: Improving Random Projections Using Marginal Information, COLT 2006

**Limitations:**

The authors address the main limitation when $\varphi_l, l \geq 2$ performs better.

I like the ideas in this paper, and future work on sketching algorithm with quantization steps could potentially build on this, but the experimental results and discussion could be more convincing.

My score is motivated by the unit circle mentioned in the questions (comparing when a one layer map is better than two layer map) which shows a small "slice" where the two layer map is better. Broadly speaking, I cannot think of any application where it is desirable to have good estimates for points extremely close to each other, yet ensure false positives do not occur (i.e. that "slice" where the Euclidean distance is $< 0.06$). I am happy to raise my score if there are potential applications, experiments or discussion (niche but realistic cases are okay).

---

> ### Author Rebuttal · Authors · 2024-08-01
>
> Weakness 1: We agree and we only claim novelty in the case $l \geq 2$; versions of the 1-layer map have been discussed in several papers. We will clarify the relationship with [6] (Charikar) and [Li et al, COLT 2006] when the maps are introduced. See overall author rebuttal for a discussion of the "niche" aspect.
>
> Weakness 2: We agree that there can be incorrect nearest neighbors after approximation. This issue seems to be inherent to any multiplicative approximation applied to nearest neighbors. Namely, given a query point and several points at nearly the same distance of the query point, any point could be returned as an approximate nearest neighbor. At the same time, because out theoretical guarantees are for multiplicative (and not just additive) error, some types of "false positives" are not possible. For example, if x is the query point and y is it's nearest neighbor at distance .05,  and there is a third point z at distance 1/2 from x, it is not possible for z to be misclassified as the nearest neighbor as long as the sketch succeeds.
>
> Weakness 3: The RCV1 dataset is available (in normalized vector form) in scikit-learn with the command from sklearn.datasets import fetch_rcv1. We also sent our code to the AC so that should be available now. Concerning the projection step, it is just for computational savings. The "true nearest neighbors" are the nearest neighbors after projection. To make the experiment simpler and more convincing, we rerun the experiment without projecting and obtained essentially the same result (see the attached pdf). We will update the experiment in the paper with the version without projection.
>
> Question 1: Our analysis shows that the additive error is small with high probability for multilayer maps and this suggests that the variance is smaller, but we do not have a formal argument for the variance.
>
> Question 2: The "good approximation" is in the sense of a small additive error with high probability. We do not need the approximation to be "unbiased." Namely, we are using  $\langle \varphi_\ell(x),\varphi_\ell(y)\rangle$  as an estimator of $f_\ell(\langle x,y \rangle )$. But we are not claiming that  $E \langle \varphi_\ell(x),\varphi_\ell(y)\rangle $ is equal to $f_\ell(\langle x,y \rangle )$. However, one of our main results (Theorem 5) shows that it is a good estimator in an additive sense as long as dimensions are large. This "additive approximation" interpretation is clarified in the next sentence (473-474).
>
> Question 3: The "true nearest neighbors" are the nearest neighbors after projection.
>
> Question 4: From Figure 1, we see that the  distance threshold at which the 2-layer map becomes better than the 1-layer map is approximately $\|x-y\| = .06$. This corresponds to $\langle x,y \rangle \approx .998 $ and $\Theta_{x,y}\approx .06$.
>
> Question 5: This is a good suggestion, we will add a conclusion in the manner suggested.
>
> Limitations:
> Please see the overall author rebuttal for a discussion of the significance of the theoretical and experimental contributions. Building on what we wrote in "Weakness 2" above, some kinds of false positives are impossible because our approximation guarantee is multiplicative and other kinds of false positives are intrinsic to any comparable kind of approximation.
>
> We agree that there is only a small "slice" where 2-layers is better, i.e. when the minimum distances is <.06. However, because we estimate distances up to a multiplicative $(1\pm\epsilon)$ error, the risk of false positives is not any larger in that scenario than it is for the datasets where minimum distances are much larger. So the question of the applicability of the 2-layer map is essentially the question of the existence of datasets where minimum distances tend to often be smaller than .06. This is rare, but it does happen. Our example is the RCV1 dataset which does have pairs of points at distance <.06. Figure 3 (and the updated experiment in the attached pdf) demonstrate that 2-layers does perform better for the 1-nearest neighbor.

---

> ### Comment · Reviewer_f7H3 · 2024-08-07
>
> Thank you for the detailed rebuttal! I am currently waiting for the AC to send out the code, as well as the global rebuttal (which I assume also has the attached pdf).

---

> ### Comment · Reviewer_f7H3 · 2024-08-07
>
> The authors have addressed my concerns, and I am happy to increase my score. I would suggest the authors to further emphasize the significance of their work (as mentioned in the global rebuttal), and maybe a discussion of false positives with respect to multiplicative error to reach out to the more applied folks.

---

> ### Author Response · Authors · 2024-08-09
>
> Thanks, we appreciate the suggestions and will plan to add some discussion of multiplicative error in the context of applications to the introduction.

---

### Official Review · Reviewer_mnzD · 2024-07-10

**Soundness:** 4
**Presentation:** 3
**Contribution:** 2
**Rating:** 4
**Confidence:** 4

**Summary:**

This paper studies the bit complexity of storing the Euclidean distance between n points X up to (1 +- eps) error.  The simplest setting assumes all points have ||x||_2 = 1, and all results depend on the "spread" of the point set m = min_{x_1,x_2 in X} ||x_1 - x_2||.

A good comparison for their results is via a JL-random project embedding; the bits required for this is:
   - O(n * (1/eps^2) log n * log(1/m eps)

The number of bits required for their approach is
  - O(n * (1/eps^2) log n * log(1/m)^{2.3})

**Strengths:**

The main advantage of this approach is that it directly records the embedding as bit vectors (well as [-1, +1] vectors, plus an implicit scaling term depending on the dimension of the data].

The paper appears technically interesting.  It introduces an idea of composing maps into bits instead of just a single map, which can show some technical improvement in theory and practice for small distances.

**Weaknesses:**

This might be of some independent interest, but as of now the improvement is quite minor.

This paper reports a very small improvement in very limited cases.
In particular, the improvement occurs when the desired error eps is much smaller than the spread parameter m; but typically for (1 +- eps) error, eps is a constant, so this seems of limited interest.

As a result, I do not think it is worth publishing at NeurIPS.

**Questions:**

none

**Limitations:**

N/A.

---

> ### Author Rebuttal · Authors · 2024-08-06
>
> Weaknesses: We believe our theoretical contributions are substantial because of the new algorithmic ideas and because our upper bound nearly matches an existing lower bound. See the overall author rebuttal for more details.

---

> > ### Comment · Reviewer_mnzD · 2024-08-08
> >
> > Thanks for the rebuttal.  I agree these results and techniques may be of purely theoretical interest, but then I do not think NeurIPS is the right venue -- at least I am not convinced.
> >
> > This is only an improvement when eps (typically a constant) is **exponentially** smaller than the spread of the point set m.  I am just not convinced this improvement is of interest to any settings relevant to the NeurIPS community.

---

> > > ### Author Response · Authors · 2024-08-09
> > >
> > > The improvement is not only an improvement when $\epsilon$ is exponentially smaller than the spread. The actual range of improvement is much better than exponential in theory and even in practice $\epsilon$ does not need to be much smaller that $m$ to see an improvement. The relevant terms to be compared are $\log(1/(m \epsilon))$ and $(\log (1/m))^{2/3}$. In theory, to have $\log(1/(m \epsilon)) \geq (\log (1/m))^{2/3}$ it is sufficient to have $1/\epsilon \geq (1/m)^{(\log(1/m))^{1.3}}$ so the threshold for improvement is no worse that quasi-polynomial (much better than exponential). This can also be verified numerically for practical values of $m$ and $\epsilon$: the largest $\epsilon$ given $m$ where $\log(1/(m \epsilon)) \geq (\log (1/m))^{2/3}$ for $m=1/4$ is $\epsilon=.48$ (so $\epsilon$ can even be larger than $m$) and for $m=1/10$ is $\epsilon=.011$.
> > >
> > > Another issue with comparing our algorithmic approach with discretized J-L is that the bound we quote for J-L is intended as an information-theoretical bound only. Our algorithm is straightforward to implement while it is not clear that there is an efficient algorithm that matches the discretized J-L bound. This is because it is based on an optimal (non-algorithmic) epsilon-net of the unit ball and it is not clear what the compression/encoding algorithm is. For each projected point, the natural compression algorithm has to find the nearest point in the (exponentially large) $\epsilon$-net.

---

> > > > ### Comment · Reviewer_mnzD · 2024-08-11
> > > >
> > > > Thank you for correcting my statement about needing eps to be exponentially smaller than m.  Still the difference needs to be more than polynomial.  The results are asymptotic so I do not find the results with small constant values compelling.  Moreover I assume normal cases of interest are eps >= 0.1 and m < 0.01.
> > > >
> > > > I agree the algorithm does appear naturally simpler.

---

### Official Review · Reviewer_4gJb · 2024-07-13

**Soundness:** 3
**Presentation:** 3
**Contribution:** 3
**Rating:** 6
**Confidence:** 3

**Summary:**

The authors investigate the bit complexity of distance preserving embeddings of points on the unit sphere and in the unit ball.
Their main finding is that iterative application of Charikar's hyperplane SimHash could use slightly fewer bits of storage than snapping a random projection to an epsilon net for certain parameters.

Existing prior work could compress a set of points to even fewer bits while preserving distances all these methods operate on the entire set. The authors' method compresses data points (vectors) individually, which is an advantage.

Empirical evaluation with synthetic and small scale real world data support the theoretical claims.

**Strengths:**

The authors study a broadly applicable problem.

The proposed method is simple to implement and comes with theoretical guarantees.

The experiments support the theory.

**Weaknesses:**

The reduction in the number of bits needed only impacts lower order factors, as explained on line 77.

Could you please discuss the many quantization methods cited on line 173 and compare yours with them both analytically and experimentally?

Figures 1 and 2 lack baselines that compress to bits, could you please add a few, including quantizing (epsilon-netting) a random projection?

**Questions:**

Could you compare the theoretical predictions of Theorem 6 with the empirical findings of Figure 1 that for distances < 0.06 two layers have lower error than one layer?

Line 340: "six time the output dimensions." Could you elaborate how 6 was chosen?

**Limitations:**

Limitations are sufficiently discussed. (Mostly) theoretical work, no negative societal implications.

---

> ### Author Rebuttal · Authors · 2024-08-06
>
> Weakness 2: Of the quantization methods discussed on line 173, the most relevant one is known as "sign random projections" [6] and some of those other papers generalize sign random projections in various ways. We will add a remark saying the our 1-layer map $\varphi_1$ is the same as the original sign random projection and that the main novelty of our work is composing multiple sign random projections.
>
> Concerning the experiments, paper [6] is essentially the same as our 1-layer map, with the same guarantees. Papers [4],[18],[21],[31] are about compressive sensing and [24] is about quantizing random Fourier features so there is no direct comparison to be made. Paper [28] provides an additive error guarantee, ours is multiplicative. Paper [10] and [23] have no theoretical estimates for the number of bits required as in our Theorem 2, instead they take the approach of bounding the variance of the estimator.
>
> Weakness 3: As our main contribution is theoretical, we did compare our technique to epsilon-netting a random projection analytically but not experimentally. Part of the difficulty is that for such an experiment it is not clear how to choose the tradeoff between the projection dimension and the size of the epsilon net (A fair experiment would have to know the optimal choice of the size of the epsilon net compared to the projection dimension.)
>
> Weakness 4: We sent the code to the AC per the instructions. We will publish the code and add a link to the paper.
>
> Question 1: According to Theorem 6, the $\ell$-layer map approximates distances up to an additive $\pm \epsilon \|x-y\|^{2-2^{-\ell+1}}$ error. Say that we use the 1-layer and 2-layer map with the same output dimension $N$. Looking at the definition of $N$ in Theorem 6 and solving that equation for $\epsilon$ we get that the 1-layer map approximates squared distances up to an additive $\frac{\sqrt{48 \log n}(\pi/\sqrt{2})\|x-y\|}{\sqrt{N}}$ error and that the 2-layer map approximates squared distances up to an additive $\frac{\sqrt{48 \log n}(\pi/\sqrt{2})^2\|x-y\|^{3/2}}{\sqrt{N}}$ error. So from this we conclude that 2-layers is better if $\|x-y\|$ is sufficiently small. However, in the context of the experiment in figure 1, Theorem 6 is not able to accurately predict exactly how small $\|x-y\|$ needs to be to make 2-layers better. The reason is that Theorem 6 assumes that $\epsilon<1- \langle x, y\rangle = \frac{\|x-y\|^2}{2} = \frac{1}{200}$ when $\|x-y\|=.05$. Theorem 6 then says that the output dimension $N$ is bigger than $\epsilon^{-2}$ so bigger than 40,000, but in the experiment in Figure 1, $N$ is only 1000. Overall, this shows that Theorem 6 is not best possible and future work could attempt to determine what one can prove without the assumption that $\epsilon<1- \langle x, y\rangle$.
>
> Question 2: We chose 6 experimentally. In particular, using 6 instead of 2 made the error significantly smaller. But, for example, using 8 or 10 instead of 6 offered no significant improvement.

---

> > ### Comment · Reviewer_4gJb · 2024-08-13
> >
> > Thanks for the explanation and clarifications.

---

### Author Rebuttal · Authors · 2024-08-01

We thank the reviewers for your constructive and helpful feedback. We have the following comments about our contributions and the experiments for all the reviewers:

The main contribution of the paper is theoretical, including algorithmic ideas and their analysis. We believe we have strong contributions and novelty there. Our contributions are about a very basic algorithmic problem: the approximation of distances up to a multiplicative $1\pm\epsilon$ error. While compared to existing upper bounds (algorithms) our improvement appears modest, it is very strong if one takes into account the known lower bounds. Namely, our algorithm's upper bound is very close to being optimal because it matches the lower bound up to the power of the $\log(1/m)$ factor. As we explain in section 1.4, it is known that for the one-way communication version of the sketching/compression problem,
$$
\Omega(\epsilon^{-2}n \log(n/\delta) \log(1/m))
$$
bits are necessary if the algorithm is to be successful with probability $1-\delta$ and $m$ is the minimum distance. Our technique uses
$$
\Theta(\epsilon^{-2}n \log(n) \log(1/m)^{2.3})
$$
bits. When $m$ is asymptotically smaller than $\epsilon$, this is closer to the lower bound than the number of bits one needs if using regular Johnson Lindenstrauss random projection which requires
$$
\Theta(\epsilon^{-2}n \log(n) \log(1/m\epsilon)).
$$
Please see the discussion in Section 1.4 for additional details.

The intent of the experiments is not to show that the proposed algorithm is an improvement in practice; it is to show that the ideas are actually implementable and have a complexity that is within the realm of other methods (i.e. it is not orders of magnitude worse). The experiments also validate the theoretical idea of "composing maps" (depth) in the following sense: They show that the 2-layer map is better than the 1-layer map for reasonable values of the parameters. To make this point stronger, we also show a real world dataset (RCV1) where the 2-layer map is better than the 1-layer map. We redid the experiment with the RCV1 dataset (i.e., Figure 3 in the paper) without first projecting the data to $\mathbb{R}^{5000}$, please see the plots in the attached PDF. The result is essentially the same as the original experiment, but is somewhat more convincing because now we are recovering the true nearest neighbors of query points in RCV1 rather than the nearest neighbors according to the projected data.

---

### Decision · Program_Chairs · 2024-09-25

**Decision:**

Accept (poster)

**Comment:**

This paper studies the problem of compressing/sketching a set of points while approximately preserving the pairwise Euclidean distances. The paper proposes a method based on the idea of ell-fold composition of a feature mapping, or a multi-layer mapping. The paper shows that the number of bits needed by the proposed sketching algorithm is theoretically smaller than what is needed by a standard JL random map in some parameter regimes. While the contribution of the paper is mainly theoretical, the proposed method is elegant and simple to implement. Some experiments are provided to validate the idea of composing maps and to show that the 2-layer map is better than the 1-layer map.

The reviewers agree that the theoretical contribution of the paper is interesting. The disagreement is basically about the practical relevance of the proposed method and, consequently, whether it is a good fit for NeurIPS. Since theory is one of the listed topics in the NeurIPS call for papers, and sketching techniques should be of interest to some part of the community, I recommend acceptance.